# Assessment of Non-Invasive Blood Pressure Prediction from PPG and rPPG Signals Using Deep Learning [note 1]

**DOI:** 10.3390/s21186022

**Published:** 2021-09-08

**Authors:** Fabian Schrumpf, Patrick Frenzel, Christoph Aust, Georg Osterhoff, Mirco Fuchs

**Affiliations:** 1Laboratory for Biosignal Processing, Leipzig University of Applied Sciences, 04317 Leipzig, Germany; patrick.frenzel@htwk-leipzig.de (P.F.); mirco.fuchs@htwk-leipzig.de (M.F.); 2Department of Orthopaedics, Trauma and Plastic Surgery, University of Leipzig Medical Center, 04103 Leipzig, Germany; christoph.aust@googlemail.com (C.A.); georg.osterhoff@medizin.uni-leipzig.de (G.O.)

**Keywords:** cuffless blood pressure, deep learning, convolutional neural network, long short-term memory, blood pressure estimation, photoplethysmogram, remote photoplethysmogram, imaging photoplethoysmogram, arterial blood pressure

## Abstract

Exploiting photoplethysmography signals (PPG) for non-invasive blood pressure (BP) measurement is interesting for various reasons. First, PPG can easily be measured using fingerclip sensors. Second, camera based approaches allow to derive remote PPG (rPPG) signals similar to PPG and therefore provide the opportunity for non-invasive measurements of BP. Various methods relying on machine learning techniques have recently been published. Performances are often reported as the mean average error (MAE) on the data which is problematic. This work aims to analyze the PPG- and rPPG based BP prediction error with respect to the underlying data distribution. First, we train established neural network (NN) architectures and derive an appropriate parameterization of input segments drawn from continuous PPG signals. Second, we use this parameterization to train NNs with a larger PPG dataset and carry out a systematic evaluation of the predicted blood pressure. The analysis revealed a strong systematic increase of the prediction error towards less frequent BP values across NN architectures. Moreover, we tested different train/test set split configurations which underpin the importance of a careful subject-aware dataset assignment to prevent overly optimistic results. Third, we use transfer learning to train the NNs for rPPG based BP prediction. The resulting performances are similar to the PPG-only case. Finally, we apply different personalization techniques and retrain our NNs with subject-specific data for both the PPG-only and rPPG case. Whilst the particular technique is less important, personalization reduces the prediction errors significantly.

## 1. Introduction

Blood pressure (BP) is regarded as an essential biomarker for various diseases. Techniques for discontinuous measurements are quite elaborated and comprise auscultatory and oscillometric cuff based methods. While they are commonly used in both clinical and home environments, they are rather unsuited for long-term measurements due to patient discomfort and potential skin irritations. Techniques for continuous BP monitoring are also readily available for some use cases and comprise arterial BP measurement and cuffless sensor solutions [1,2,3]. The former is invasive and thus limited to clinical settings; the latter requires the use of multiple sensors, e.g., electrocardiogram (ECG) electrodes and PPG sensors. They actually allow continuous monitoring but are still uncomfortable for patients. Long-term measurements even require regular recalibration using an additional cuff. Other but usually less practicable techniques to measure BP comprise ultrasound, tactile sensor based approaches, and vascular unloading based methods [4].

In recent years research mainly focused on BP estimation from cuffless multi and single sensor solutions. The former often utilize time or phase differences between different signals (usually ECG and PPG or multiple PPG) related to the blood volume propagation through arteries [5,6,7,8,9]. The latter mainly exploit morphological properties of blood volume dynamics derived from PPG measurements on a particular site [10,11,12,13,14,15,16,17,18,19,20]. PPG-only based methods are particularly interesting. Not only do they target single sensor solutions, but rather is their underlying signal generation principle very much similar to that of camera based techniques known as rPPG. If PPG signals could be utilized for BP estimation, a fully contactless method like rPPG might be feasible as well, thus providing opportunities for a lot of clinical and non-clinical application scenarios. In fact, some studies have investigated the estimation of BP [21,22,23] or its correlates [24,25,26,27] from these signals.

Recent progress in biomedical engineering is mainly driven by machine learning (ML) techniques and recently by the advent of deep learning methods. Such methods are commonly applied to biomedical image processing but also to emotion recognition as well as in the field of cardiac arrhythmia detection [28,29,30]. Deep learning based BP prediction methods typically pursue feature based approaches or perform learning in an end-to-end manner. Feature based methods exploit spectral or temporal properties of the PPG signal which are fed into a learning algorithm to predict systolic and diastolic BP. End-to-end procedures leverage the waveforms themselves and implicitly derive features to predict BP. Often the accuracies of these methods were reported to be in line with well-established standards (BHS or AAMI) [31]. It is, however, common to only report an indicator for the mean performance of an algorithm without sufficiently taking the underlying data distributions into account. This is problematic since the performance of a learning algorithm tends to be biased towards the mode of this distribution. That means the error appears to be small for the majority of samples but is much larger when deviating towards the tails of their distribution. To the best of our knowledge, only one recent work studied the distribution of the BP prediction error [32]. They reported an insufficient accuracy of the investigated method with respect to the full BP range but only divided the BP range coarsely into three intervals. This underpins that a more detailed analysis of the error distribution is critical for the assessment of BP prediction methods in particular for clinical applications.

The BP distribution of a dataset is affected by various issues. The variability of the pulse morphology among subjects, e.g., caused by age and cardiovascular diseases, certainly affects the association of the signal shape to a particular BP. The equipment in a hospital setting differs as well and, more importantly, the contact pressure of a PPG sensor can affect the pulse morphology [33,34,35]. Therefore the ability of the learning algorithm to generalize well may be impaired. On the individual scale of a subject, the BP variation is often limited during the measurement, e.g., due to medication (often not reported in databases), limited physical activity or due to short record periods. This may also affect the data distribution, particularly when training, validation and test sets are not split carefully. Lastly, the data used to train a learning model is often not publicly available. The question which subjects were used for training and testing when using publicly available databases remains frequently unanswered. Hence, it is challenging to assess whether an improved prediction error results from methodological improvements or just from data selection. While these are issues that arise for both PPG and rPPG based methods, recent works disputed the usefulness of rPPG signals for BP estimation altogether [36,37].

This paper targets: (1) an empirical evaluation of the parameterization of input signals (e.g., segment cropping from the continuous signals) suited for both PPG and rPPG based BP prediction with established neural network (NN) architectures; (2) a detailed assessment of PPG based BP prediction performance on sufficiently small intervals of the systolic and diastolic targets and an analysis of the impact of different dataset creation strategies on the prediction error; (3) the effect of personalization on networks trained on PPG data; (4) rPPG based BP prediction on a dataset recorded in a clinical setting based on a pre-trained and fine-tuned network; (5) the effect of personalization by fine-tuning networks using subject-specific data. The results of this paper add an in-depth analysis of PPG and rPPG based end-to-end machine learning approaches to BP prediction to the state of knowledge in the field of non invasive patient monitoring. We compare the prediction performances of several established neural architectures with the results of recently published studies. Based on these comparisons, we derive requirements for datasets and training pipelines in order to achieve an unbiased assessment of a BP predictor’s performance with regard to clinical applications. Finally, we extend the investigated end-to-end methods to rPPG based BP predictions and assess the feasibility of camera based BP prediction using transfer learning.

This work represents an extension of the paper presented at CVPR 2021 [38]. We added recently published work related to the topic. Furthermore, we trained neural networks with different datasets that were created using sample based and subject based division. The goal was to investigate the effect of mixing subject-specific data between training, validation and test set on the prediction error. To extend the scope of our analysis we added a long short-term memory (LSTM) network as a fourth neural architecture. Finally, we analyzed different personalization strategies for PPG and rPPG based blood pressure prediction.

## 2. Related Work

### 2.1. Deriving BP Using Parameterized Models

Early works for cuffless BP estimation utilize the pulse transit (PTT) or pulse arrival time (PAT). The PTT is the time delay for the pulse to travel between two different arterial sites and the PAT is the time delay between the electrical onset (R-peak in the ECG) and the arrival of the evoked pulse wave at a particular site [39]. They can be used to derive the pulse wave velocity (PWV) using the Moens-Korteweg equation [40,41]. Gesche et al. used the PWV and inferred BP values by means of a linear regression model [42]. This method resulted in a commercially available smartwatch for sleep research (SOMNOtouch NIBP, SOMNOMEDICS GmbH). Socrates et al. [9] validated this device and found a standard error of 4.2 mmHg for systolic BP (SBP) and diastolic BP (DBP) during sleep and awake phases.

### 2.2. BP Prediction Using PPG Features

Haddad et al. used morphological features of the PPG-waveform to classify BP into normal and hypertonic ranges using multi-linear regression [18]. Others employed dense neural networks (DNN) to derive BP from time based morphological features such as pulse width, pulse amplitudes and heart rate. They also extracted features from the first and second derivative of the PPG waveform [13,15,19]. Other authors used recurrent neural networks (RNN) to derive BP from time- and frequency based PPG-features [5,14,43]. In [6,7], the authors trained a very deep RNN by introducing skip connections between layers to overcome the vanishing gradient problem [44]. Yang et al. constructed time series from morphological PPG features and PTT values. They divided the time series into high- and low-frequency components and fed them into a DNN and RNN for BP estimation [8].

### 2.3. End-to-End Approaches to Predict BP

Slapničar et al. [10] used a parallel architecture consisting of three residual neural networks (ResNet) to predict BP using the PPG waveform and its first and second derivatives. They also used a subject based calibration resulting in a significantly reduced mean average error (MAE). Schlesinger et al. used a siamese convolutional neural network (CNN) to predict BP variations with respect to a calibration value. They used PPG spectrograms as inputs for their NN [12]. Baek et al. utilized ECG and PPG to derive time and frequency domain input segments and proposed a new BP prediction model based on multiple losses [45]. Sadrawi et al. used an autoencoder structure consisting of a LeNet and U-net architecture to predict ABP waveforms. They employed a genetic algorithm to select an optimal model among several training runs. The U-net autoencoder performed best with an MAE of 3.26 mmHg and 1.91 mmHg for SBP and DBP, respectively [46].

Recent studies [47,48,49] used CNNs in combination with RNNs to estimate BP. The features vectors are derived from PPG based CNN embeddings and fed into an RNN for BP prediction. Eom et al. [48] achieved an MAE of 0.06 mmHg and 5.42 mmHg for systolic and diastolic BP using this approach. Wang et al. used the PPG time-course and first and second-order derivatives as input to a parallel structure consisting of an LSTM and a CNN. They classified BP into five different classes and achieved an overall accuracy of 91% [50]. A more straightforward approach was pursued by Han et al. who used a CNN with PPG-waveforms as input and achieved an overall F1-score of 0.9 when classifying BP into different hypertension classes [20]. Xing et al. used a simple DNN for BP estimation. They transformed the PPG signal into the frequency domain and used its amplitude and phase spectrum as input. The mean error based on the MIMIC-II database was 0.06 mmHg and 0.01 mmHg for systolic and diastolic BP [11]. Lee et al. used a bidirectional LSTM followed by a multilayer perceptron to estimate BP values based on PPG, ECG and the balistocardiogram. They recruited 18 subjects and achieved an MAE of 2.62 mmHg and 2.03 mmHg for SBP and DBP after leave-one-out cross validation. After fine tuning the final dense layer of their model using 20% of their test subject’s data they were able to improve those results further [51]. A similar approach was used by Mou et al. investigating different combinations of DNN, CNN and LSTM. Their best performing model was a LSTM followed by a CNN. After training using three subjects from the MIMIC database they achieved an MAE of 2.37 mmHg and 4.42 mmHg for SBP and DBP, respectively [52]. Harfiya et al. created an LSTM based autoencoder for sequence-to-sequence learning. They first trained encoder and decoder stages to predict PPG signals from PPG signals downloaded from the MIMIC-II database. Finally, they used transfer learning by fine tuning the decoder to predict ABP waveforms. Their overall MAE was 5.04 mmHg and 2.41 mmHg for SBP and DBP [5].

## 3. Materials and Methods

### 3.1. Datasets

#### 3.1.1. PPG Data

The MIMIC-III database consists of thousands of records from various hospitals collected between 2001 and 2008 and sampled at 25 Hz [53]. We used a subset of the MIMIC-III database available on Kaggle. It consists of 12,000 records of PPG, ECG and ABP signals. Their authors applied extensive preprocessing on the waveforms to provide a clean and valid dataset [54,55]. It represents only a small fraction of the MIMIC-III database. but is therefore compact and contains signals with acceptable quality. However, the subject affiliation is unknown thus rendering it unsuited for evaluating the model performance. Hence, we only used this dataset in the first part of our work to evaluate the parameterization of input signals. It is denoted as MIMIC-A for the remainder of this paper.

A much larger portion of the MIMIC-III database was downloaded using scripts provided by [10], resulting in a total pool of 4000 records and approximately 150 million samples (PPG and ABP signal pairs). This dataset is used for performance evaluations and will be referred to as MIMIC-B for the remainder of this paper. To ensure a balanced dataset we limited the contribution of each subject to 2000 samples when creating training, validation and test sets from this sample pool. Imbalanced datasets can lead to an overestimation of the predictor’s performance. In our case, imbalance arises if certain subjects contribute substantially more samples to the dataset than others, especially if these subjects appear in the training as well as in the validation and test set. To investigate whether this is the case with our dataset we analyzed the number of samples for every subject in our population.

#### 3.1.2. rPPG Data

Data for camera based BP prediction was recorded in a study at the Leipzig University Hospital. The study design was approved by the ethics committee of the University of Leipzig. Subjects were informed about the content of the study and gave written consent prior to taking part. 50 subjects with a planned surgical intervention were enrolled. After surgery, the patients were transferred to the intensive care unit. Our recording system consisted of an industrial USB camera (IDS UI-3040CP, 32 fps) connected to a PC (Intel NUC NUC7I7BNH). Videos of the subjects face and upper body with approximately two hours duration were recorded. Ground truth BP was derived from the bedside monitor with one minute temporal resolution.

### 3.2. Neural Network Architectures

We used four different neural network architectures to predict BP values. The first is AlexNet which is a CNN architecture originally developed for image classification [55]. We adopted its structure in order to use PPG time series as inputs and return systolic and diastolic BP values instead of class predictions.

Second, we used a ResNet, i.e., a very deep CNN architecture [44]. Their skip connections efficiently account for the vanishing gradient problem occuring in deep architectures. The ResNet was modified in the same manner as the AlexNet. The dimensions for the input layer of these networks were ( Nsamp × 1) in the univariate case and (Nsamp × 3) when using the raw time series and its first and second order derivatives. The final classification in each original model was replaced by a regression layer consisting of two neurons for SBP and DBP with a linear activation function.

Third, we used the architecture published by Slapničar et al. [10]. It consists of a spectrotemporal network with residual connections. It is a parallel structure and processes PPG signals and their first and second-order derivatives.

Lastly, we implemented a bidirectional LSTM model as a recurrent neural network architecture that uses PPG signals as input and outputs systolic and diastolic blood pressure [56]. LSTM can detect long-term dependencies in input sequences by introducing feedback connections and are frequently being used in natural language processing. In the context of blood pressure estimation we used them to detect temporal dependencies within the PPG time series that are correlated with blood pressure. Our LSTM architecture consists of one convolutional layer with ReLU activation as the first layer, followed by four LSTM layers and one densely connected layer. A schematic of the neural architecture can be seen in Figure 1. The convolutional layer consisted of 32 filters with kernel size 5 and strides 1. The ReLU function was used as activation. The LSTM layers consisted of 64, 64 and 32 hidden unity, respectively. Finally, the dense layer consisted of 128 neurons. The size of all layers were selected empirically.

### 3.3. PPG Signal Processing

We first aimed at studying the effect of the signal length on the prediction performance. We divided the PPG and ABP signals from the MIMIC-A dataset into segments of different lengths, i.e., 1, 2, 5, 7, 9, 11, 13, 15, 17 and 20 s. This dataset is called const_time_xx. Since this approach leads to an interruption of single PPG cycles at the beginning and the end of each time window, we created an alternative dataset to evaluate whether this affects the prediction performance. Therefore, we ensured that only full cycles (i.e., complete beats) are contained in a time window under test using the following procedure. We estimated the heart rate based on the PPG by detecting the spectral component with the highest amplitude. Next, we divided the signal into time segments containing an integer number of 1, 2, 5, 7, 9, 11, 13, 15, 17 and 20 PPG-waves. The heart reate varies across subjects and so does the number of samples within these segments since they were constructed from integer multiples of PPG-waves. In order to achieve feature vectors of equal length, we determined the duration of each PPG wave as one second or 125 samples since the sampling frequency of the MIMIC-III database was 125 Hz. We then resampled each window to a total of $N_{P}\times 125$ samples (e.g., 500 samples for a window containing $N_{P}=4$ PPG waves) achieving an effective heart rate of 60 bpm. This way, the number of beats could be held constant while also ensuring a constant length of all windows belonging to a certain number of ppg waves. It is obvious that this eliminates absolute temporal information since the sampling frequency now varies among PPG windows. The resulting dataset is called const_beats_xx. We processed ABP signals in a similar manner to yield datasets consisting of PPG-ABP pairs.

Recent studies showed that, apart from the raw PPG itself, derivatives yield useful information on the cardiovascular state [57] and can be useful for BP as well [10,15,50]. Hence, we employed the first and second order derivatives of each PPG-window as well and studied whether this multivariate approach reduces the BP prediction error.

We derived the ground truth systolic and diastolic BP from the ABP segments. We used a peak detection algorithm to detect systolic and diastolic peaks [58,59] and derived the reference BP as the median of all peaks within each segment. We employed several plausibility checks. All BP values outside a physiologically plausible range of 75 to 165 mmHg and 40 to 80 mmHg for systolic and diastolic BP, respectively, were discarded. Median heart rates in each window that exceeded the ranges of 50 to 140 bpm were also rejected.

### 3.4. Evaluation of NN Input Sequences

We first trained our NNs with the MIMIC-A dataset to determine the proper cropping strategy and window length. We used 100,000 randomly drawn samples for training the model and additional 25,000 samples for validating and testing, respectively. The neural architectures and the training pipelines were implemented using Google Tensorflow 2.4 and Python 3.5 (Adam optimizer, α = 0.001, 50 epochs, mean squared error loss). The models with the lowest validation loss were used for subsequent performance tests.

We conducted three repeated training procedures for both AlexNet and ResNet for each length of the input time segment under test. We then employed a paired t-test to evaluate whether interrupting PPG cycles when cropping segments from the continuous PPG signal affects the MAE. We conducted this analysis separately for the AlexNet and ResNet architecture.

To determine the optimal window length we employed a twofold strategy. First, we evaluated the NN performances with respect to the input length of the segments derived from PPG signals. Second, we evaluated how a particular length would affect the signal-to-noise ratio (SNR) of rPPG segments. We aimed at selecting a length that, on the one hand, ensures high SNR values (thus resulting in longer segments) but, on the other hand, maximizes the number of segments (and therefore enforcing shorter segments) available for NN training. The latter is motivated by the fact that various studies have shown that a large number of training examples is crucial for the successful training of NNs. Hence, our procedure aimed at finding a trade-off resulting in a sufficiently large number of training examples with an acceptable SNR for rPPG, given the limited amount of samples in the rPPG dataset. As before, we divided the rPPG dataset into windows of different length and calculated the SNR for every time window. We used a threshold of −7 dB SNR above which a rPPG segment would be accepted. We finally used a the shortest window length that provides a large number of acceptable time windows while still being long enough to expect good results for BP prediction with the NN.

The SNR has been calculated by using a method proposed by de Haan et al. [60]. We detected the spectral peak representing the pulse rate and calculated the energy $E_{P}$ within a frequency band around this peak and its first harmonic. We also calculated the spectral energy $E_{S}$ of the remaining spectrum excluding the the frequency bands around the peak related to the pulse and itsa first harmonic. The SNR was calculated using Equation (Equation 1).
(1)$$SNR=10\log_{10}\frac{E_{P}}{E_{S}}$$

### 3.5. PPG Based Prediction

#### 3.5.1. Data Preprocessing

The MIMIC-B dataset was divided into windows using the optimal cropping strategy and window length determined in Section 3.3. Since this data came directly from the physionet.org database, additional preprocessing steps had to be applied. First, a 4th order Butterworth band-pass filter was applied to the PPG-signals. Cut-off frequencies were set to 0.5 Hz and 8 Hz. Second, the signal-to-noise ratio (SNR) was calculated for all signals [60]. All signal windows with an SNR below −7 dB were discarded. All PPG-windows were normalized to zero mean and unit variance.

#### 3.5.2. NN Training and Validation

The datasets were split into training, validation and test sets on a subject-basis to prevent contamination of the validation and test set by training data. We used 3750 subjects for training and 625 subjects for validation and testing. Among these subjects, we have randomly drawn 1,000,000 samples for training, 250,000 samples for validation and 250,000 samples for testing.

In a second experiment, we randomly selected 750 subjects from our sample pool to create the dataset. Each of these subjects contributed 2000 samples. In contrast to the first experiment, the dataset was split randomly into training, validation and test set. In this scenario, the neural network might see samples from subjects during validation and testing whose data were also in the training set since subject affiliation was not taken into account during the split. The goal was to evaluate the difference in performance between mixed and non-mixed datasets.

Input pipelines and NNs (AlexNet, ResNet, model from Slapničar et al. and LSTM) were implemented using Google TensorFlow 2.4.1 and Python 3.8 was used for training (Adam optimizer, α§ = 0.001, euclidian loss). Training was stopped either after 200 epochs or if the validation loss did not improve for 10 epochs. We used the models with the lowest MAE on the validation set for predictions on the test set.

For evaluation purposes, we additionally used a mean regressor that always predicts the systolic and diastolic BP from the training set. A well generalizing ML method will exceed the mean regressor’s performance. We used the MAE metric to assess the performance of all methods. In contrast to other work, we determined the prediction errors both for the full dataset and separately for rather narrow BP bins, altogether spanning the range of ground truth BP values contained in the datasets.

#### 3.5.3. PPG Based Personalization

Personalization can improve the blood pressure predictor’s performance [10]. We randomly drew 20 subjects from the non-mixed testset and used a part of each subject’s data to fine-tune the pre-trained neural architecture from Section 3.5.2. To compare different personalization schemes we first used 20% of the test subject’s data by randomly drawing samples. In a second setting we used the first 20% of the test subject’s data. To ensure comparability between the test results we used the same test data for every personalization strategy. We reserved half of the last 80% of every test subject’s data for testing while the remaining half could be used to draw training data for the random personalization.

This is a more realistic scenario insofar as a patient might undergo a cuff based measurement when admitted to the hospital which serves as calibration for a non-invasive method. The goal of this analysis was to investigate whether performance on the test subject could be improved by personalization and if a certain personalization strategy may be beneficial.

### 3.6. rPPG Based Prediction

#### 3.6.1. Preprocessing

ROIs on a subject’s forehead and cheeks were labelled manually after the recording using custom software. We used the Plane-orthogonal-to-skin (POS) algorithm to derive the pulse wave from the skin pixels [61]. The resulting rPPG-signal was then inspected visually. Twenty-five subjects with heavy motion artifacts, frequent movement or insufficient lighting were deemed unsuitable and excluded from further analysis. Remaining data was divided into windows based on their heart rate. Seven heartbeats were included (compare Section 3.3 and Section 3.4) and each window was resampled according to the procedure for the MIMIC-B dataset (Section 3.5.1). We calculated the SNR value for each rPPG window according to [60] and excluded windows with an SNR below −7 dB. Ground truth BP values were downloaded from the bedside monitor.

#### 3.6.2. Transfer Learning

The resulting dataset was used to fine-tune the pre-trained NNs. Transfer learning exploits the idea that rPPG and PPG waveforms share similar properties and should therefore give rise to similar relevant features during NN training. Due to the low amount of data, however, an entire retraining of these NNs was not feasible. We therefore only optimized the final layer while freezing all other network weights. We used the Adam optimizer (α = 0.001) and fine-tuned until the MAE stopped decreasing. The best model given by leave-one-out cross-validation was then used to evaluate the model using the test subject. We also investigated whether the personalization strategy presented in [10] improves the prediction accuracy. In addition to the training dataset we reserved 20% of the test subject’s data for training and validated the model with the remaining 80%. We performed two training runs for each test subject. For the first training we drew the additional training data randomly from the test subject’s data. In the second training the first 20% of the test subject’s data were included in the training.

## 4. Results

### 4.1. PPG Based Prediction

#### 4.1.1. Input Signals

Figure 2 shows the training results of the AlexNet and ResNet architectures with the const_HR and const_time datasets with and without additional time derivatives. Note that we combined the results obtained for all variations of the segment lengths for this analysis. The prediction errors based on const_HR are lower in comparison to const_time. The significance of these findings was confirmed using a paired t-Test (*p* < 0.01). The use of the first and second order derivatives shown as const_HR_derivative and const_time_derivative did not yield to a general improvement compared to the univariate case. Just the SBP MAE of the AlexNet was slightly lower than in the univariate case. Due to this and for the sake of simplicity, we did not consider derivatives for our further analysis.

According to Section 3.4, we aimed to derive a suitable length of the input signal as a trade-off from PPG and rPPG data. First, we evaluated the prediction error with respect to PPG input signals. We expected a slight increase of the error towards longer segments since the morphology of PPG cycles slightly varies over time and therefore would introduce undesired ambiguities with respect to the underlying BP. However, our empirical analysis (three repetitions for each NN and length parameter) did not confirm this effect and resulted in an almost equal prediction error for each tested length. Note that because of the high computational effort that would be necessary to obtain a sufficiently high number of repeated training procedures for each NN and length parameter (30+ each), a statistical justification of this effect can not be provided. From this perspective, the longest possible segment length (20 s) seemed useful.

Second, we analyzed the SNR of rPPG segments with respect to their lengths (Figure 3) and aimed to maximize the number of resulting samples available for training. It should be noted that the SNR measure becomes less appropriate towards small segment lengths which also explains the sharp decline at the beginning of the curve. It seemed reasonable to select a value beyond the bend of the curve to account for this effect. We finally selected 7 s as the segment length for all further analyses.

#### 4.1.2. MIMIC-B Dataset

We downloaded 4000 subjects off the MIMIC-III dataset resulting in a pool of approximately 170 Mio. samples. Figure 4 shows a histogram of the number of subjects that contribute a certain amount of samples to this pool. It can be seen that the number of samples per subject spans a broad range from just a few hundred to over 500,000. This may lead to an imbalanced dataset if certain subjects contribute a large number of samples compared to other subjects with just a few samples. As a consequence, the neural network’s performance may be overestimated especially if the data of the dominating subjects would split between training, validation and test set. To counteract this, we limited the contribution of each subject to the dataset to 2000 samples ensuring a balanced dataset. This represents a tradeoff between a high number of subjects in the dataset and a sufficient number of samples per subject while maintaining a reasonable dataset overall size.

Figure 5 shows the BP distribution of the training, validation and test set for the mixed and non-mixed dataset created according to the methods described in Section 3.5.2. It can be seen that the distributions are highly similar and span an equal range of systolic and diastolic blood pressure. This makes them equivalent for training in terms of their distribution. However, the non-mixed dataset applies the additional constraint that subjects may be either training, validation or test subjects avoiding mixing subjects between training, validation and test set.

#### 4.1.3. Predicting BP Using PPG Data

The neural architectures were trained with the mixed and non-mixed datasets. We compared their performances in terms of MAE, thereby also employing the mean regressor as a baseline. A Kolmogorov-Smirnov Test (KS-Test) was used to test for statistical significance (α = 0.05). The results on the test set can be seen in Figure 6 and Table 1. There is a statistically significant decrease in MAE among the neural architectures when using the mixed dataset for training instead of the non-mixed dataset. Differences between mixed and non-mixed dataset when using the mean regressor were also statistically significant but much less pronounced compared to the neural architectures. Comparing the results on the non-mixed dataset among all methods the neural architectures achieve a significantly lower MAE compared to the mean regressor. However, the improvement is not large enough to be of any practical relevance. The ResNet achieved the lowest MAE on the mixed and non-mixed dataset with 16.4 mmHg and 8.5 mmHg (non-mixed) as well as 7.7 mmHg and 4.4 mmHg (mixed) for SBP and DBP, respectively.

To analyze the BP-dependence of the MAE we divided the acceptable BP input range into bins of 10 mmHg width and calculated the error for each bin separately for each architecture trained on the MIMIC-B dataset. The results are shown in Figure 7. The error strongly varies across BP bins when training on the non-mixed dataset. This emphasizes a large dependence of the error on the underlying BP. Each architecture achieves the lowest MAE in a range of 100–130 mmHg (SBP) and 50–70 mmHg (DBP). The BP distribution of the test set is shown at the bottom of Figure 6. It can be seen that the error is inversely proportional to the number of samples in the respective bin. This suggests that the ML methods achieve a training loss reduction by concentrating their predictions on the BP range with the most frequent training examples. The neural networks achieve a lower MAE than the mean regressor suggesting that the predictions are based on actual morphological features learned from the input data. However, the large errors at the edge of the BP distribution render the underlying neural networks inadequate for clinical applications. The MAE based on the mixed dataset shows a much less pronounced dependence on the BP range. We have seen a significant improvement when using the mixed dataset compared to the non-mixed dataset especially at the edges of the BP distributions.

#### 4.1.4. PPG Based Personalization

To investigate the effect of personalization on the ppg-based training we randomly drew 20 subjects from the test set of the non-mixed dataset. We performed the personalization for each test subject separately and used 20% of the test subject’s data for fine tuning. We used two strategies to select this subset. First, we used 20% of the samples at the very beginning of each measurement and second we used 20% randomly drawn samples from the complete measurement. We evaluated the statistical significance of the observed prediction results using a two sample KS-Test (α = 0.05).

Figure 8 shows the results of the personalization for test subjects 1 to 10. Figure A1 shows the results for test subjects 11 to 20. It is evident that personalization decreases the MAE substantially. Personalization had the biggest effect when using the AlexNet architecture and the model from Slapničar et al. where 50% of all test subjects showed a significant improvement of the MAE after fine tuning the network. However, the choice of the personalization strategy had only limited effect on the MAE. When fine tuning the ResNet, only 4 of 20 of the Y test subjects showed significantly different MAE distributions between the different personalization strategies. AlexNet and the model of Slapničar et al. were even less affected with 5% and 0% of the test subjects showing significant differences in MAE distributions, respectively.

### 4.2. rPPG Based Prediction

Next, we fine-tuned the NNs trained on PPG data for BP prediction using rPPG data and a leave-two-out cross validation scheme. Similar to the ppg based fine tuning we performed three training scenarios. In the first training, we used 15 of the 17 subjects for fine tuning the neural network on rPPG data while the remaining two were used for validation and testing, respectively. In the second training we added the first 20% of the measurement samples of the subject under test to the training set which corresponds to the first personalization strategy introduced before. In the third training, the 20% randomly drawn samples from the whole measurement of the test subject were included in the fine tuning data. As before, the complementary set of the conjunction of these two subsets were used for testing.

Table 2 (top) shows the overall MAE when predicting BP values using the test subject and the PPG pre-trained networks without any rPPG fine tuning. It can be seen that the MAE is much higher than the PPG based performance as observed in Section 4.1.2. It can further be seen in Table 2 that fine tuning the networks using rPPG data improved the SBP MAE considerably, reducing it from 28.9 mmHg to 14.1 mmHg when adapting the ResNet architecture.

Figure 9 shows the overall MAE distribution of all test subjects. Fine tuning the networks using rPPG data led to a significant improvement of the MAE especially regarding the SBP but also for DBP. Personalizing the training with additional data from the test subject yielded a moderate improvement of the MAE for some architectures. The overall MAE of the ResNet after personalization including the first 20% of the test’s subject data resulted in an SBP MAE of 12.7 mmHg compared to 14.1 mmHg without personalization. Differences in MAE between fine tuning strategies were statistically significant for all architectures. Only the AlexNet showed no significant difference between fine tuning with and without personalization including the first 20% of the test subject’s data.

Figure A2 and Figure A3 show the MAE for each of the subjects included in the analysis that showed an improvement in MAE after fine tuning using rPPG data. Fine tuning with or without personalization resulted in an improved SBP MAE for 13 subjects. Another four subjects showed no improvement. Their results are depicted in Figure A4 and Figure A5, The comparison between BP predictions with and without fine tuning resulted in a substantial improvement in MAE for most subjects especially regarding the SBP-MAE. The improvement in DBP MAE was less pronounced mostly due to the smaller value range of the diastolic blood pressure. These results imply a significant difference in morphology between PPG and rPPG and underline the importance of retraining when adapting neural architectures to camera-based signals.

## 5. Discussion

This paper investigated the feasibility of BP estimation using PPG- and rPPG-based pulse wave signals. Importantly, our aim was not to derive a particularly accurate model to achieve state-of-the-art performance for BP prediction from PPG but rather: (1) to explore the dependence of these models on some important time domain properties of the PPG and rPPG input signal, (2) to learn how these models perform not only in terms of a mean performance on a given dataset but on a more fine grained scale of multiple BP bins and (3) to investigate the feasibility of transfer learning from PPG for rPPG based BP prediction. We also ensured a careful split of the data and divided the dataset based on subjects to avoid contaminating the validation and test data with training data.

First, we conducted an empirical evaluation of the parameterization of the input signals that were suited to train our NNs. In particular, we analyzed the window length, cropping of segments from continuous signals and the use of derivatives. We used established NN architectures (i.e., AlexNet and ResNet) from the literature and adopted them for BP prediction. Moreover, we used the architecture as presented by Slapničar et al. which is optimized for BP prediction from PPG. We found that the use of derivatives does not provide significant improvements and is less important. Avoiding the introduction of strong phase discontinuities using an appropriate cropping of segments from continuous signals significantly improves the prediction performance. Finally, the total length of the input sequence was less important with respect to the PPG-based prediction errors. The selection of the length of segments was mainly driven by rPPG data to maximize the number of samples in the dataset.

Second, our analysis of the BP range dependent prediction error reveals that the NNs are partly superior over the mean regressor. However, effect sizes were small. Besides that, when training with the non-mixed dataset, we found a strong dependence of the bin-wise prediction error on the number of samples in the particular BP bin in the underlying distribution. The most accurate predictions occurred in BP bins containing the most samples. This emphasizes the tendency towards predicting the mode of the training dataset. A similar dependence was found in [32], which led to the retraction of a publicly available smartphone app for BP prediction. However, their results are based on a much coarser subdivision of the BP range.

In order to compare our results to previous work, we additionally evaluated the mean performance. None of the results met the requirements as defined in the relevant BHS and AAMI standards, which require the probability of a BP measurement device to provide an acceptable error (BP < 10 mmHg) to exceed 85% [31]. Especially the high MAEs in the higher and lower BP ranges pose a problem since many clinical applications rely on an acceptable accuracy in hypo- and hypertensive ranges. Importantly, our BP MAE is in accordance with Slapničar et al. [10] who also accounted for subject specific affiliations to training and testing datasets. Note that we did not employ any hyperparameter tuning which might further improve our results, but certainly only on a gradual scale. In contrast, other authors who did not explicitly mention a subject-based dataset split reported substantially lower prediction errors [33,34,35,62].

To analyze this effect we created a PPG dataset, were individual subjects were plit across training, validation and test sets. When using this mixed dataset, the dependence of the MAE on the BP range was strongly reduced resulting in an improved MAE especially at the tails of the blood pressure distribution. This confirms the initial hypothesis that testing the neural networks using data from subjects seen previously during training leads to an overestimation of the neural network’s performance. In a practical application, data from new subjects (e.g., from patients admitted to hospital) would lead to a prediction error much closer to the MAE reported on the non-mixed dataset rendering the whole method unsuited for clinical use. The impact of age, comorbidities, medication and measurement equipment have a large influence on PPG morphology [34,63]. These differences can be exploited as features for the non-invasive assessment of cardiovascular aging [64,65]. However, our results suggest that such differences in PPG morphology prevent the investigated NNs from generalizing well.

Recent works by Zhang et al. addressed this issue using neural architectures specifically tailored to learn domain-invariant features [66]. This was beyond the scope of our study since we focused on NN architectures which are already applied on a broad basis. Division of the subjects into age groups and training the models for each group individually could pose a way to increase the predicition performance. However, such information was not available in the MIMIC-III database. Datasets used for training neural networks for BP prediction should also include demographic information on the subjects inlcuded. The results emphasize the importance of inter-individial differences in the PPG signals and that it is of course not sufficient to ensure that any data samples must only be used once either in the training set or in the test set but rather to use all the data from a single subject only for training or testing in order to obtain a well generalizing model, unless some sort of personalization is intended.

While some authors reported that they accounted for such a subject-dependent split (e.g., [6,8,10,12,20]) others did not and our results suggest that in this case a particularly high prediction performance might partly result from potential train/test split violations. However, it should also be noted that such a careful split is not always possible when using publicly available databases like the MIMIC-II subset where subject specific information is often not provided. The signal segments are not assigned to specific subjects and it might even be the case that data from multiple subjects appear in the same segment. This is due to some limitations in the recording procedure in real clinical environments where bedside monitors are used to measure data from several patients one after another before storing the (raw) data.

In order to investigate the clinical applicability further we applied personalization to both the PPG and rPPG based blood pressure prediction methods. Personalization comprised the usage of parts of the test subjects’ data selected randomly or systematically for training. We observed an improvement of the BP prediction error for all investigated neural architectures and for PPG as well as for rPPG based training compared to the training without personalization. Finally we analyzed whether specific personalization strategies had any effect on the MAE. We found no clear difference between the strategies. However, calibration of the neural networks using subject-specific data prior to any BP prediction might be a viable way to improve the accuracy of (r)PPG based blood pressure prediction.

During our video recordings at the University of Leipzig Medical Center, variations in lighting as well as frequent movements of the subjects were very common. As a result, the derived rPPG signals were affected by artifacts rendering part of the recordings unusable for BP estimation. Since patients cannot be restricted in their freedom of movement, alternative approaches to rPPG measurement have to be taken into consideration. Recording patients using IR cameras has proven beneficial in reducing the influence of ambient lighting [67]. Furthermore, using an end-to-end method for rPPG derivation could lead to a more stable signal since frequent movement complicates the placement of ROIs on the subject’s face. Yu et al. published PhysNet, a neural network that derives rPPG signals from face videos directly [68]. Using this method it would be possible to totally dispense of ROI placement since the whole face can be used to estimate the rPPG signals possibly leading to a cleaner signal.

## 6. Conclusions

Our study emphasizes the following: To develop an ML technique that aims to be of relevance for practical clinical applications, one must: (1) evaluate the prediction error of a model over the full systolic and diastolic BP range and (2) therefore carefully take the data distributions in the training and test sets into account. While this is of course obvious and an important rule for designing ML algorithms in general, it is not yet treated with the necessary care among published literature for ML based BP prediction. In particular, differences in age, the health state and medications are likely to have a strong influence on the BP distribution in a dataset. Therefore, an appropriate split into subject-dependent training and test sets must be ensured in any case. Including data of training subjects in the test set by randomly splitting the datasets on a sample-basis leads to an overestimation of the model performance. Our results suggest that some of the previously reported BP prediction performances might be related to a violation of this aspect

It is also worth to further investigate the relation between the PPG morphology and the sensor contact pressure which might have an even more severe effect on the ability of an ML model to generalize at all. This is especially the case when using public databases where patient records can emerge from various sources. Given our findings, it does not seem unlikely that the mentioned and probably even more issues causing variations in the PPG morphology might render BP predictions from PPG a highly ill-posed problem for real world applications.

Third, we fine-tuned our NNs using rPPG data collected in a clinical study. We found that the estimation error greatly varied between subjects. We emphasize that we had only limited training data for fine-tuning and therefore could only fine-tune the final layer of each NN. Given more data, it would be conceivable to tune additional layers and possibly enhance the prediction performance.

Similar to rPPG, we also investigated the effect of personalization on the networks trained with PPG data. We discovered a considerably improved prediction accuracy in comparison to the mean regressor.

Remote measurement of BP using standard RGB cameras is still an active field of research. Given the results of our study, it seems very questionable if not even impossible that algorithms solely relying on morphological properties of PPG-only signals are suited for the derivation BP measurements especially in the light of the strong requirements that have to be met for clinical applications. BP estimation based on rPPG signals and their morphological properties appears to be even more challenging with respect to these requirements.

## Figures and Tables

**Figure 1 sensors-21-06022-f001:**
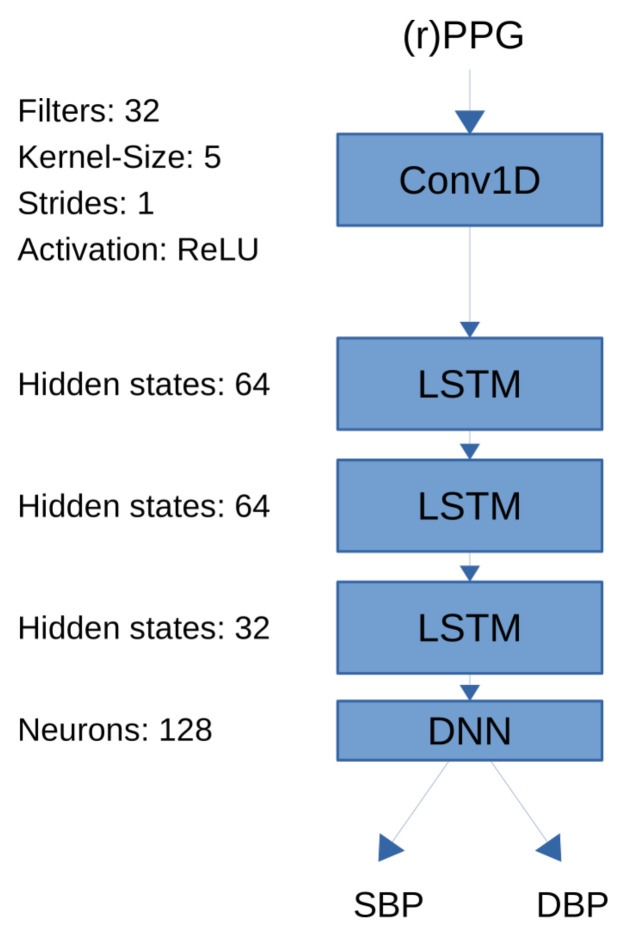
Long short-term memory (LSTM) network architecture used for remote photoplethymographic (rPPG) and photoplethysmographic (PPG) based training. The model consists of one convolutional layer followd by three long short term memory (LSTM) layers followed by one dense layer to predict systolic blood pressure (SBP) and diastolic blood pressure (DBP).

**Figure 2 sensors-21-06022-f002:**
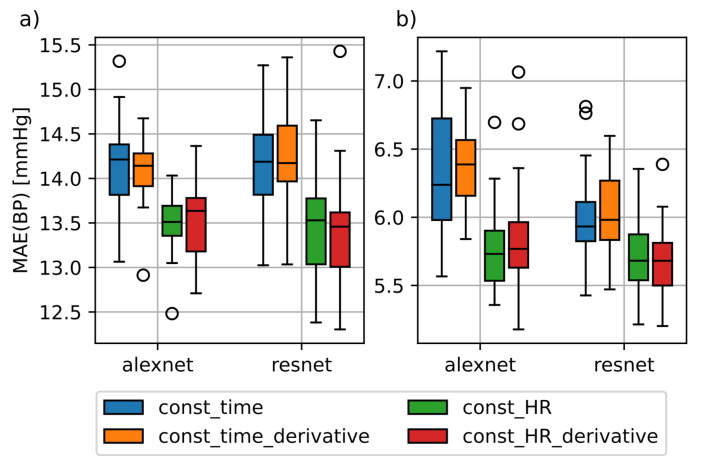
Difference in mean absolute error (MAE) between const_time and const_HR dataset for AlexNet and ResNet. (**a**): MAE for the systolic blood pressure; (**b**): MAE for diastolic blood pressure; const_time_derivative and const_HR_derivative datasets include the first and second order derivatives of the photoplethysmography (PPG) windows contained in the datasets const_time and const_HR.

**Figure 3 sensors-21-06022-f003:**
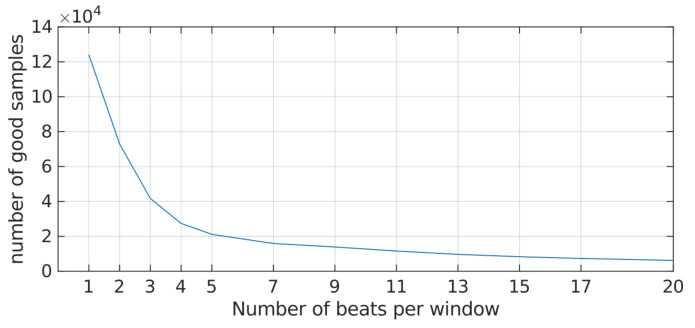
Fraction of acceptable samples in the remote photoplethymography (rPPG) dataset with respect to the window length. Samples with a signal-to-noise ratio (SNR) below the threshold of −7 were discarded.

**Figure 4 sensors-21-06022-f004:**
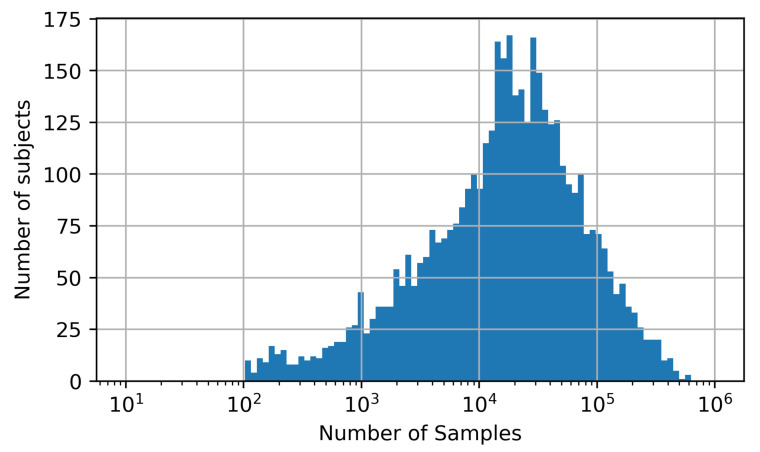
Histogram of the number of subjects contributing a certain amount of samples to the data pool from which training, validation and test sets were created.

**Figure 5 sensors-21-06022-f005:**
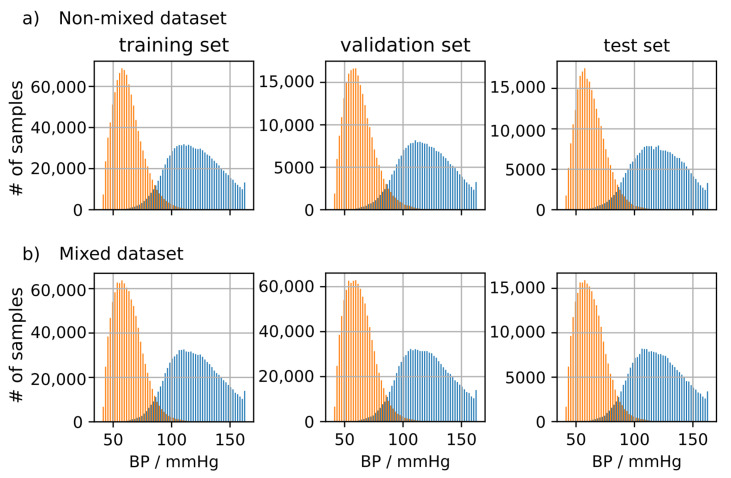
(**a**): Distribution of systolic (SBP) and diastolic blood pressure (DBP) in the non-mixed dataset. Samples were selected ensuring strict separation of subjects between datasets. (**b**) Distribution of SBP and DBP in the mixed dataset. Samples were selected randomly disregarding subject affiliations.

**Figure 6 sensors-21-06022-f006:**
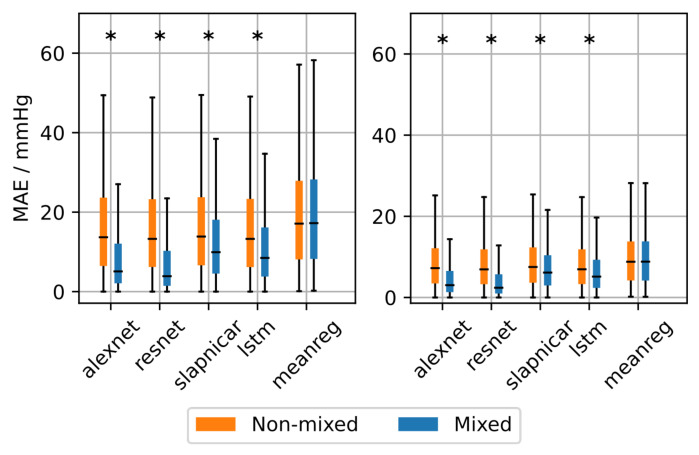
Overall mean absolute error (MAE) after training the neural architectures with the mixed and non-mixed dataset. Asterisks denote statistical significant differences between MAE distributions (checked using a Kolmogorov-Smirnov test).

**Figure 7 sensors-21-06022-f007:**
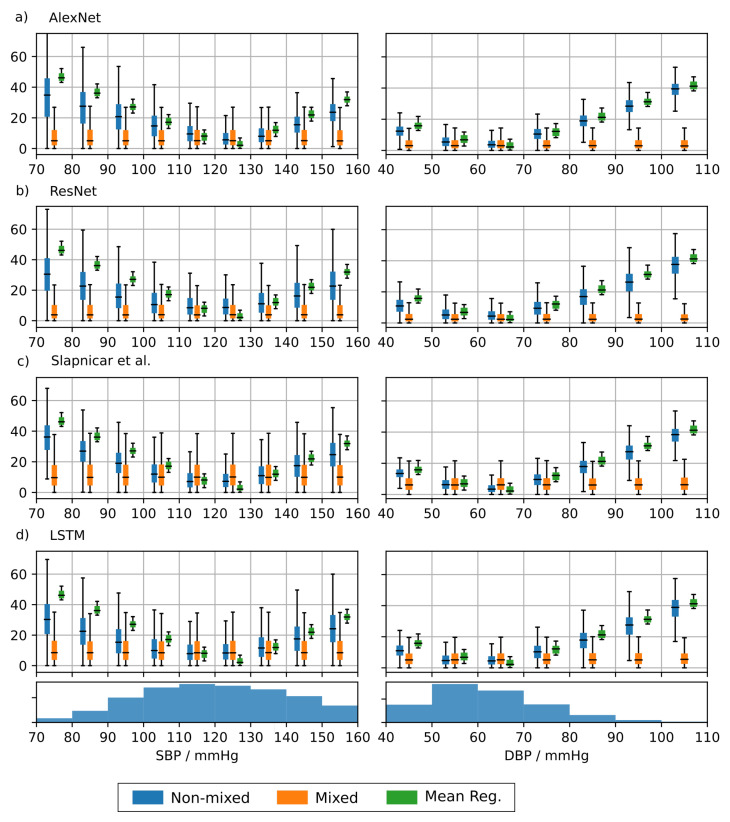
Mean absolute error (MAE) depending on the arterial blood pressure (ABP) based blood pressure. The admissible blood pressure range was divided into bins of width 10 mmHg. For reference, the distribution of systolic (SBP) and diastolic (DBP) blood pressure in the training set is displayed on the bottom of the figure. Results are shown for AlexNet (**a**), ResNet (**b**), architecture by Slapničar et al. (**c**) and LSTM (**d**).

**Figure 8 sensors-21-06022-f008:**
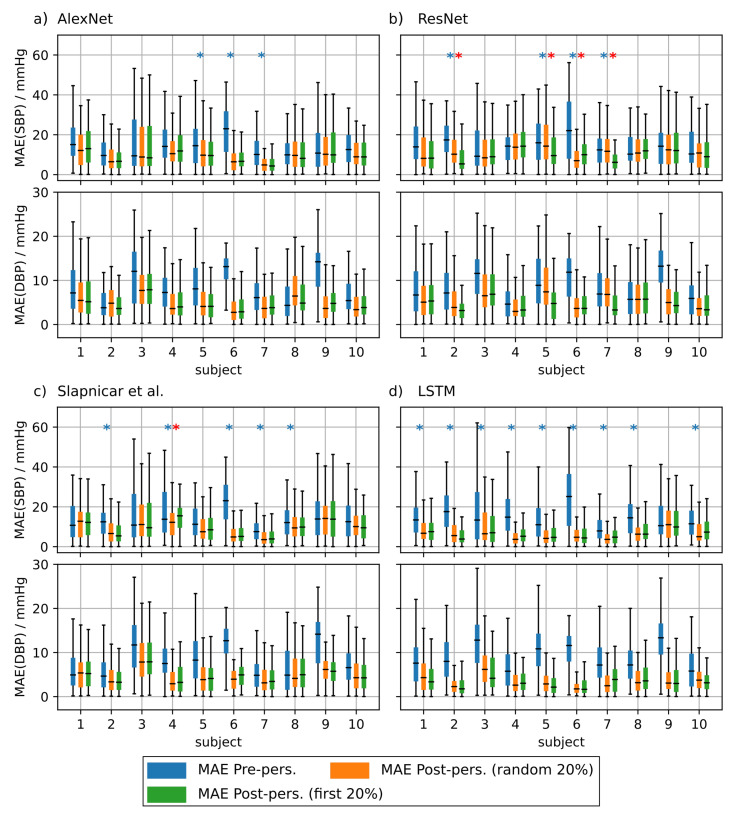
Mean absolute error (MAE) of the blood pressure (BP) prediction using additional test subjects without personalization (blue boxes), with personalization using randomly drawn training data (orange boxes) and personalization using the first 20% of the test subject’s data for training (green boxes). Blue asterisks denote statistical significant differences between MAE with and without personalization. Red asterisks denote statistical significant differences in MAE between personalization strategies (Kolmogorov–Smirnov test). Only 10 of the 20 tested subjects are depicted in this figure. Results are shown for AlexNet (**a**), ResNet (**b**), architecture by Slapničar et al. (**c**) and LSTM (**d**).

**Figure 9 sensors-21-06022-f009:**
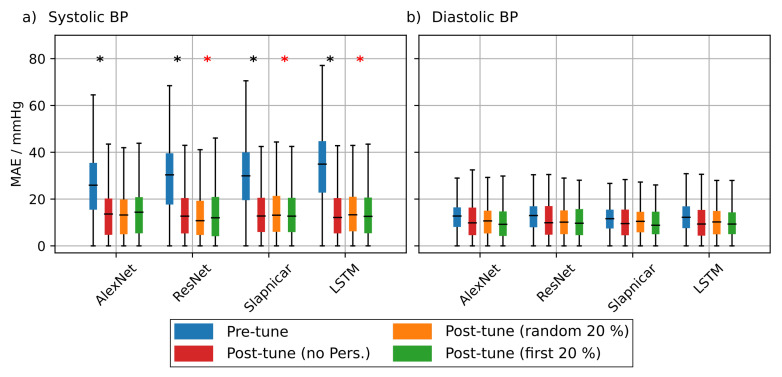
Comparison of the overall mean absolute error (MAE) after evaluation of the pretrained model using photoplethymographic (PPG) data without fine tuning (blue boxes), after fine tuning (red boxes), after fine tuning with personalization using randomly drawn data (orange boxes) and after personalization using the first 20% of the test subject’s data. Statistical significant differences of the MAE pre and post fine tuning are marked with a black asterisk. Statistical significant differences in MAE among the two personalization strategies are marked with a red asterisk (checked using a Kolmogorov-Smirnov test). (**a**,**b**) show the results for SBP and DBP respectively.

**Table 1 sensors-21-06022-t001:** Mean absolute error (MAE) in mmHg for mixed and non-mixed (n.m.) trainings as well as fine tuning using subjects from the test set with and without personalization. Results after fine tuning and personalization have not been compared to the mean regressor.

	Dataset	Architecture				
		**AlexNet**	**ResNet**	**Slapničar**	**LSTM**	**Mean-Reg**
SBP	Mixed	8.8	7.7	12.9	11.6	19.6
	Non-mixed	16.6	16.4	16.8	16.4	19.6
	pre pers. (n.m.)	15.8	16.2	15.2	15.7	-
	pers. rand (n.m.)	11.8	13.0	10.8	8.5	-
	pers. first (n.m.)	12.2	12.3	11.1	9.0	-
DBP	Mixed	4.9	4.4	7.5	6.7	9.9
	Non-mixed	8.7	8.5	8.8	8.6	9.8
	pre pers. (n.m.)	10.1	9.8	9.8	9.9	-
	pers. rand (n.m.)	6.0	6.3	5.8	4.5	-
	pers. first (n.m.)	6.1	5.8	5.9	4.6	-

**Table 2 sensors-21-06022-t002:** Mean absolute error (MAE) values when predicting systolic (SBP) and diastolic (DBP) blood pressure values using rPPG data and the neural networks pre-trained with PPG data. The networks werw evaluated before and after rPPG-based fine tuning. Furthermore, two different personalization strategies were analyzed.

	MAE(SBP) [mmHg]	MAE(DBP) [mmHg]
Before fine tuning		
AlexNet	28.1	13.8
ResNet	28.9	13.3
Slapničar	29.6	11.5
LSTM	33.5	12.4
After fine tuning w/o personalization		
AlexNet	14.0	11.0
ResNet	14.1	11.2
Slapničar	14.8	10.3
LSTM	13.6	10.3
After fine tuning with personalization (first 20%)		
AlexNet	14.2	10.7
ResNet	12.7	10.8
Slapničar	15.2	10.5
LSTM	14.4	10.5
After fine tuning with personalization (random 20%)		
AlexNet	14.0	11.0
ResNet	14.1	11.2
Slapničar	14.8	10.3
LSTM	13.6	10.3

## Data Availability

The source code used for generating the presented results is publicly available on GitHub, https://github.com/Fabian-Sc85/non-invasive-bp-estimation-using-deep-learning (accessed on 21 July 2021).

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
