# Peer review of "Assessment of Non-Invasive Blood Pressure Prediction from PPG and rPPG Signals Using Deep Learning†"

_sensors, 2021, doi:10.3390/s21186022_

Round 1
Reviewer 1 Report
1. The following statements in the conclusions are strange: "While it still seems questionable to derive BP from PPG-only data, it remains even more questionable whether rPPG data is actually suitable for BP estimation [33,34,62]. Future studies should concentrate on end-to-end approaches since the rPPG- signal’s low SNR hampers methods based on morphological features, especially when different skin tones, motion and changes in illumination are involved [63,64]." After having done an 18 pages long paper on this subject, this contradicts the stated objectives. In my opinion, this study becomes now of minimal relevance, being more in the category "art for art".
2. As an unwritten rule of writing articles, no references should be placed in the conclusions. Please remove the references from the conclusions.
3. Please explicitly mention the novelty degree of the paper and how it was achieved.
4. Reference number 35 is missing the year. Please rectify.
5. There are too many references (a total of 64). Some of them are superfluous and too many even for a reader specialist in the field. Please remove approximately 30% of them and leave the truly essential ones.
Reviewer 2 Report
In their manuscript, the authors extensively analyzed the feasibility of the parameterization of input signals for both photoplethysmography (PPG) and remote PPG (rPPG) based blood pressure (BP) prediction with established neural network architectures; moreover, they assessed the PPG and rPPG based BP prediction performance and the effect of personalization on networks.
The topic is of great interest, especially for its clinical application.
The paper is well written and both the results and the discussion are clear and complete.
I suggest that the following points should be addressed before consideration for acceptance and publication:
- Some typing errors should be corrected, for example, line 94, “the scope of your analysis” probably stands for “the scope of our analysis”.
- The authors should describe the abbreviations mentioned in every table and figure in the appropriate legend.
- The applications of PPG and neural network other than BP assessment should be briefly mentioned in the introduction, for example, in the setting of arrhythmias and state anxiety (Pereira T, et al. Photoplethysmography based atrial fibrillation detection: a review. NPJ Digit Med. 2020 Jan 10;3:3. doi: 10.1038/s41746-019-0207-9; Perpetuini D, et al. Prediction of state anxiety by machine learning applied to photoplethysmography data. PeerJ. 2021 Jan 15;9:e10448. doi: 10.7717/peerj.10448).
Reviewer 3 Report
The paper entitled “Assessment of non-invasive blood pressure prediction from PPG and rPPG signals using deep learning” reports about the capability to estimate blood pressure form PPG signals using deep learning approaches. The paper is very interesting and well written. The topic is relevant considering the spread of wearable and contactless PPG devices for cardiovascular monitoring. In my opinion, before publication, some concerns need to be addressed:
- Page 6 lines 231-236: It is not clear to me what the Authors mean with “The new sampling frequency was chosen in a way that the PPG-window contains a heart rate of 60 bpm”. Do they mean that the temporal resolution and the amplitude of the window would allow to take into account heart rate of 60 bpm. Please better specify this aspect. Moreover, the Authors stated that all the segments were resampled to have the same length, losing temporal information. Did the Authors try to not downsample the data? Obviously, it could be computationally challenging, but maybe for a reduced number of PPG waves (e.g. 5) it could be possible.
- The Authors stated that the age and the cardiovascular state of the patients could affect the performance of the BP estimation. In fact, PPG waveform is deeply influenced by the age and the vascular condition. Did the Authors try to split the dataset in age classes and define different models for each age class? Maybe the performances of the model could increase. Please discuss this aspect in the Discussion section. Please refer to:
- Chiarelli, A. M., Bianco, F., Perpetuini, D., Bucciarelli, V., Filippini, C., Cardone, D., ... & Merla, A. (2019). Data-driven assessment of cardiovascular ageing through multisite photoplethysmography and electrocardiography. Medical engineering & physics, 73, 39-50.
- Dall’Olio, L., Curti, N., Remondini, D., Harb, Y. S., Asselbergs, F. W., Castellani, G., & Uh, H. W. (2020). Prediction of vascular aging based on smartphone acquired PPG signals. Scientific reports, 10(1), 1-10.
- Please specify how the SNR of the signals was computed.
- The Authors stated that “Twenty-five subjects with heavy motion artifacts, frequent movement or insufficient lighting were deemed unsuitable and excluded from further analysis”. Together with the performances that does not meet the BHS and AAMI standards, the lighting and the movement of the subject could be a challenging issue to introduce this method of continuous monitoring in clinical practice. Please, provide in the Discussion section some possible solutions to overcome this limitation. Maybe employing infrared cameras and considering also different ROIs.
- Figure 5. Please add the y label. The mixed and not-mixed distributions seem to be equal, is it right?
